# Duplex Sequence-to-Sequence Learning for Reversible Machine Translation

**Zaixiang Zheng**[1,2]*, **Hao Zhou**[2], **Shujian Huang**[1], **Jiajun Chen**[1], **Jingjing Xu**[2], **Lei Li**[3]*
[1]National Key Laboratory for Novel Software Technology, Nanjing University
[2]ByteDance AI Lab    [3]UC Santa Barbara
{zhengzaixiang,zhouhao.nlp,xujingjing.melody}@bytedance.com
{huangsj,chenjj}@nju.edu.cn, lilei@ucsb.edu

## Abstract

Sequence-to-sequence learning naturally has two directions. How to effectively utilize supervision signals from both directions? Existing approaches either require two separate models, or a multitask-learned model but with inferior performance. In this paper, we propose REDER (**RE**versible **D**uplex Transform**ER**), a parameter-efficient model and apply it to machine translation. Either end of REDER can simultaneously input and output a distinct language. Thus REDER enables *reversible machine translation* by simply flipping the input and output ends. Experiments verify that REDER achieves the first success of reversible machine translation, which helps outperform its multitask-trained baselines up to 1.3 BLEU. [1]

## 1  Introduction

Neural sequence-to-sequence (*seq2seq*) learning [Sutskever et al., 2014] has been extensively used in various applications of natural language processing. Standard seq2seq neural networks usually employ the *encoder-decoder* framework, which includes an encoder to acquire the representations from the source side, and a decoder to yield the target side outputs from the encoded source representation [Bahdanau et al., 2015, Gehring et al., 2017, Vaswani et al., 2017].

Generally, given paired training data $\mathcal{D}_{xy} = \mathcal{X} \times \mathcal{Y}$, where $\mathcal{X}$ is the source side and $\mathcal{Y}$ is target side of the corresponding seq2seq task, supervision signals are always *bidirectional*. Thus we can learn not only the mapping from source to target ($f_{\theta_{xy}} : \mathcal{X} \mapsto \mathcal{Y}$) but also the mapping from target to source ($f_{\theta_{yx}} : \mathcal{Y} \mapsto \mathcal{X}$). This is very common in many applications. For example, given parallel corpus, we can obtain machine translation models from both Chinese to English and English to Chinese, or transfer between different stylized texts [Yang et al., 2018, He et al., 2019].

Typical seq2seq learning utilizes the bidirectional supervisions by splitting the bidirectional supervisions into two unidirectional ones and trains two individual seq2seq models on them, respectively. However, such splitting ignores the internal consistencies between the bidirectional supervisions. Thus how to make better use of the bidirectional supervisions remains an open problem. One potential solution is multitask learning [Johnson et al., 2017], which jointly leverages the bidirectional supervisions within one model and expects the two unidirectional supervisions could boost each other (Figure 1(b)). But it does not work well in our case due to the parameter interference problem [Arivazhagan et al., 2019, Zhang et al., 2021]. For instance, in the setting of multitask Chinese-to-English and English-to-Chinese bidirectional translations, the encoder/decoder of seq2seq networks is trained to simultaneously understand/generate both Chinese and English, which always results in performance

---

*Work was done when Zaixiang Zheng was a final-year PhD candidate at Nanjing University and an intern (now FTE) at ByteDance AI Lab; and when Lei Li was also at ByteDance AI Lab.

[1]Code is available at `https://github.com/zhengzx-nlp/REDER`.

drop due to the divergent nature of the two languages. Another branch of solutions lies in cycle training [Sennrich et al., 2016, He et al., 2016, Zheng et al., 2020], which still deploys two individual models for separately learning, but explicitly regularize their outputs by the cycle consistency of seq2seq problems. Such approaches usually need to introduce extra monolingual data in machine translation.

In this paper, we propose to explore an alternative approach for utilizing bidirectional supervisions called *duplex sequence-to-sequence* learning. We argue that current seq2seq networks do not benefit from multitask training because the seq2seq networks are *simplex* from the view of telecommunications[2]. Specifically, the data stream only flows from encoder to decoder in the current seq2seq learning, and such simplex property makes the multitask learning suffer from the parameter interfere problem. Instead, if we have *duplex* neural networks, where the data stream can flow from both ends, each of which only specializes in one language when learning from bidirectional supervisions (Figure 1(c)), thus potentially alleviating the parameter interference problem. As a result, the bidirectional translation can be achieved as a reversible and unified process. In addition, we could still incorporate cycle training in the duplex networks using only one single model.

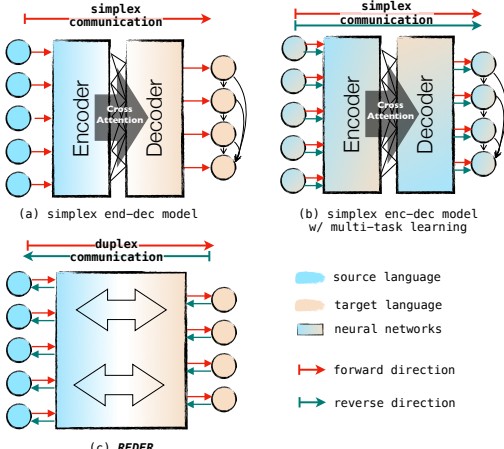

Figure 1: Illustration of different sequence-to-sequence neural models in regards to modeling direction and generation formulations.

**Definition 1.** *A sequence-to-sequence neural network with parameter $\theta$ is duplex if it satisfies the following: its network has two ends, i.e., a source end and a target end; both source and target ends can take input and output sequences; its network defines a forward mapping function $f_\theta^\rightarrow : \mathcal{X} \mapsto \mathcal{Y}$, and a reverse mapping function $f_\theta^\leftarrow : \mathcal{Y} \mapsto \mathcal{X}$, that satisfies the following reversibility: $f_\theta^\leftarrow = (f_\theta^\rightarrow)^{-1}$ and $f_\theta^\rightarrow = (f_\theta^\leftarrow)^{-1}$; besides, it satisfies the following cycle consistency: $\forall \boldsymbol{x} \in \mathcal{X} : f_\theta^\leftarrow (f_\theta^\rightarrow(\boldsymbol{x})) = \boldsymbol{x}$ and $\forall \boldsymbol{y} \in \mathcal{Y} : f_\theta^\rightarrow (f_\theta^\leftarrow(\boldsymbol{y})) = \boldsymbol{y}$.*

Based on the idea of duplex network, we propose REDER[3], the **RE**versible **D**uplex Transform**ER**, and apply it to machine translation. Note that, building duplex seq2seq networks is non-trivial: a) vanilla encoder-decoder network is irreversible. The output end of the decoder cannot take in input signals to exhibit the encoding functionality and vice versa; b) the topologies of the encoder and the decoder are heterogeneous, *i.e.*, the decoder works autoregressively, while the encoder is non-autoregressive. To this end, we therefore design REDER without explicit encoder and decoder division, in which we introduce the reversible Transformer layer [Gomez et al., 2017] and fully non-autoregressive modeling to solve the above two problems respectively. As a result, REDER works in the duplex fashion, which could better exploit the bidirectional supervisions for achieving better downstream task performance.

Experimental results show that the duplex idea indeed works: Overall, REDER achieves BLEU scores of 27.50 and 31.25 on standard WMT14 EN-DE and DE-EN benchmarks, respectively, which are top results among non-autoregressive machine translation models. REDER achieves significant gains (+1.3 BLEU) compared to its simplex baseline, whereas multitask learning does hurt the translation performance of simplex models both in the autoregressive (-0.5 BLEU) and non-autoregressive settings (-1.3 BLEU). Although REDER adopts fully non-autoregressive modeling to realize the duplex networks, the gap of BLEU between REDER and autoregressive Transformer is negligible, meanwhile REDER can directly translate between two directions and enjoys $5.5\times$ inference speedup as a bonus of non-autoregressive modeling. To our best knowledge, REDER is the first duplex seq2seq network, which enables the first feasible reversible machine translation system.

---

[2]In telecommunications and computer networking, the simplex communication means the communication channel is unidirectional while the duplex communication is bidirectional.

[3]The model's name is a *palindrome*, which implies the model works from both ends.

## 2 REDER for Reversible Machine Translation

In this section, we introduce how to design a duplex neural seq2seq model, the REDER (**RE**versible **D**uplex Transform**ER**), that satisfies Definition 1, and its application in machine translation that realizes the first neural reversible machine translation system.

### 2.1 Challenges of Reversible Machine Translation

Reversible natural language processing [Franck, 1992, Strzalkowski, 1993] and its applications in machine translation [van Noord, 1990] were proposed for the purpose of building machine models that understand and generate natural languages as a reversible, unified process. Such process resembles the mechanism of the ability that allows human beings to communicate with each other via natural languages [Franck, 1992]. Despite the success of neural machine translation with deep learning, designing neural architectures for reversible machine translation yet remains under-studied and has the following challenges:

*Reversibility*. Typical encoder-decoder networks and their neural components, such as Transformer layers, are irreversible, *i.e.*, one cannot just obtain its inverse function by flipping the same encoder-decoder network. To meet our expectation, an inverse function of the network should be derived from the network itself.
*Homogeneity*. Intuitively, a pair of forward and reverse translation directions should resemble a homogeneous process of understanding and generation. However, typical encoder-decoder networks certainly do not meet such computational homogeneity due to extra cross attention layers in the decoder; and also because of the discrepancy that the decoder works autoregressively but the encoder does non-autoregressively. To meet our expectations, the separation of encoder and decoder should be no more exist in the desired network.

### 2.2 The Architecture of REDER

To solve the above challenges, we include two corresponding solutions in REDER to address the reversibility and homogeneity issues respectively, *i.e.*, the Reversible Duplex Transformer layers, and the symmetric network architecture without the encoder-decoder framework.

Figure 2 shows the overall architecture of REDER. As illustrated, REDER has two ends: the source end (left) and the target end (right). $\theta$ is the model parameter, shared by both directions. The architecture of REDER is composed of a series of identical Reversible Duplex Transformer layers. When performing the source-to-target mapping $f_\theta^\rightarrow$, a source sentence $x$ (blue circles) 1) first transforms to its embedding $e(x)$ and enters the source end; 2) then goes through the entire stacked layers and evolves to final representations $\mathbf{H}_L$ which predicts probabilities; 3) finally its target translation (orange circles) will be generated from the target ends. The generation process is fully non-autoregressive.

Likewise, the target-to-source mapping $f_\theta^\leftarrow$ is achieved by reversely executing the architecture of REDER from target end to source end. We will dive into the details of the key components of REDER in the following parts.

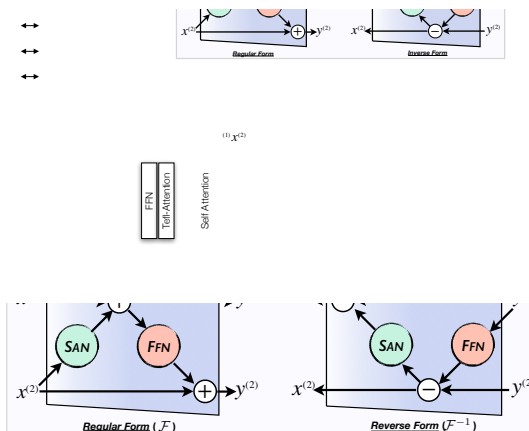

Figure 2: The proposed REDER for duplex sequence-to-sequence generation. The bottom two diagrams show the computation of the regular and reverse forms of a reversible layer. Notice that, to make the whole model symmetric, we reverse the 1-th to $L/2$-th layers, such that the overall computational operations of forward and reverse of REDER are homogeneous.

**Reversibility: Reversible Duplex Transformer layers.** We adopt the idea of the reversible residual network (RevNet, Gomez et al., 2017, Kitaev et al., 2020) in the design of the reversible duplex Transformer layer. The bottom of Figure 2 illustrates the forward and reverse computations of a layer. Each layer is composed of a multi-head self-attention (SAN) and a feed-forward network (SAN) with

a special reversible design to ensure duplex behavior, where the input and output representations of such a layer are split by 2 halves, *i.e.*, $[\mathbf{H}_{l-1}^{(1)}; \mathbf{H}_{l}^{(2)}]$ and $[\mathbf{H}_{l-1}^{(1)}; \mathbf{H}_{l}^{(2)}]$. Formally, the *regular form* of the $l$-th layer $\mathcal{F}_l$ performs as follow:

$$\mathbf{H}_l = \mathcal{F}_l(\mathbf{H}_{l-1}) \Leftrightarrow [\mathbf{H}_l^{(1)}; \mathbf{H}_l^{(2)}] = \mathcal{F}_l([\mathbf{H}_{l-1}^{(1)}; \mathbf{H}_{l-1}^{(2)}]),$$

$$\text{where} \quad \mathbf{H}_l^{(1)} = \mathbf{H}_{l-1}^{(1)} + \text{SAN}(\mathbf{H}_{l-1}^{(2)}), \quad \mathbf{H}_l^{(2)} = \mathbf{H}_{l-1}^{(2)} + \text{FFN}(\mathbf{H}_l^{(1)}).$$

The *reverse form* of $\mathcal{F}_l^{-1}$ can be computed by subtracting (instead of adding) the residuals:

$$\mathbf{H}_{l-1} = \mathcal{F}_l^{-1}(\mathbf{H}_l) \Leftrightarrow [\mathbf{H}_{l-1}^{(1)}; \mathbf{H}_{l-1}^{(2)}] = \mathcal{F}_l^{-1}([\mathbf{H}_l^{(1)}; \mathbf{H}_l^{(2)}]),$$

$$\text{where} \quad \mathbf{H}_{l-1}^{(2)} = \mathbf{H}_l^{(2)} - \text{FFN}(\mathbf{H}_l^{(1)}), \quad \mathbf{H}_{l-1}^{(1)} = \mathbf{H}_l^{(1)} - \text{SAN}(\mathbf{H}_{l-1}^{(2)}).$$

For better modeling the reordering between source and target languages, we employ relative self-attention [Shaw et al., 2018] instead of the original one [Vaswani et al., 2017].

**Homogeneity: Symmetric network architecture without encoder-decoder framework.** To meet our need to ensure homogeneous network computations for forward and reverse directional tasks, we therefore choose to discard the encoder-decoder paradigm.

*Symmetric network.* To achieve homogeneous computations, one solution is to make our network symmetric, as depicted at the top of Figure 2. Specifically, we let the 1-th to $L/2$-th layers be the reverse form, whereas the latter $(L/2 + 1)$-th to $L$-th be the regular form:

$$\overrightarrow{f_\theta}(\boldsymbol{x}) \triangleq \mathcal{F}_1^{-1} \circ \cdots \circ \mathcal{F}_{L/2}^{-1} \circ \mathcal{F}_{L/2+1} \circ \cdots \circ \mathcal{F}_L(\boldsymbol{x}),$$

$$\overleftarrow{f_\theta}(\boldsymbol{y}) \triangleq \mathcal{F}_L \circ \cdots \circ \mathcal{F}_{L/2+1} \circ \mathcal{F}_{L/2}^{-1} \circ \cdots \circ \mathcal{F}_1^{-1}(\boldsymbol{y}),$$

where $\circ$ means a layer is connected to the next layer. Thereby the forward and reverse computations of REDER become homogeneous: the forward computational operation series reads as a palindrome string $\langle \texttt{fs} \cdots \texttt{fssf} \cdots \texttt{sf} \rangle$ and so does the reverse series, where $\texttt{s}$ and $\texttt{f}$ denote SAN and FFN.

*Fully non-autoregressive modeling.* Note that without encoder-decoder division, REDER works in a fully non-autoregressive fashion in both reading and generating sequences. Specifically, given an input sequence $\boldsymbol{x}$, $\mathbf{H}_{0,i} = [\mathbf{H}_{0,i}^{(1)}; \mathbf{H}_{0,i}^{(2)}] = [\boldsymbol{e}(x_i); \boldsymbol{e}(x_i)]$, is the $i$-th element of REDER's input, which is the concatenation of two copies of the embedding of $x_i$. Once a forward computation is done, the concatenation of the output of the model $[\mathbf{H}_{L,i}^{(1)}; \mathbf{H}_{L,i}^{(2)}]$ serves as the representations of target translation. And then, a softmax operation is performed to measure the similarity between the model output $[\mathbf{H}_{L,i}^{(1)}; \mathbf{H}_{L,i}^{(2)}]$ and the concatenated embedding of ground-truth reference $[\boldsymbol{e}(y_i), \boldsymbol{e}(y_i)]$, to obtain the prediction probability:

$$p(y_i|\boldsymbol{x}; \theta) = \text{softmax}([\boldsymbol{e}(y_i); \boldsymbol{e}(y_i)]^\top [\mathbf{H}_{L,i}^{(1)}; \mathbf{H}_{L,i}^{(2)}]/2).$$

We can likewise derive the procedure of $\overleftarrow{f_\theta}$ for the target-to-source direction. Due to the conditional independence assumption among target tokens introduced by non-autoregressive generation, the log-likelihood of a translation becomes:

$$\log p(\boldsymbol{y}|\boldsymbol{x}; \theta) = \sum_i \log p_\theta(y_i|\boldsymbol{x})$$

**Modeling variable-length input and output.** Encoder-decoder models can easily model variable-length input and output of most seq2seq problems. However, discarding encoder-decoder separation imposes a new challenge: the width of all the layers of the network is depending on the length of the input, thus it is very difficult to allow variable-length input and output, especially when the input is shorter than the output. We resort to the Connectionist Temporal Classification (CTC) [Graves et al., 2006] to solve this problem, a latent alignment approach with superior performance and the flexibility of variable length prediction. Given the conditional independence assumption, CTC is capable of efficiently finding all valid alignments $\boldsymbol{a}$ which derives from the target $\boldsymbol{y}$ by allowing consecutive repetitions and inserting *blank* tokens, and marginalizes log-likelihood:

$$\log p_{\text{ctc}}(\boldsymbol{y}|\boldsymbol{x}; \theta) = \log \sum_{\boldsymbol{a} \in \Gamma(\boldsymbol{y})} p_\theta(\boldsymbol{a}|\boldsymbol{x}),$$

where $\Gamma^{-1}(\boldsymbol{a})$ is the collapse function that recovers the target sequence by collapsing consecutive repeated tokens, and then removing all blank tokens. Note that CTC requires that the length of source input should not be smaller than the target output, which is not the case in machine translation. To deal with this, we follow previous useful practice by upsampling the source tokens by 2 times [Saharia et al., 2020, Libovický and Helcl, 2018], and filter those examples when the target lengths are still larger than the one of upsampled source sentences.

**Remark.** *Reversibility in REDER can be assured in the continuous representation level, where REDER can recover from output representations (last layer) to input embeddings (first layer), which is also the motivation and basis of the auxiliary learning signal, i.e.* $\mathcal{L}_{\text{fba}}$, *in the next section. Reversibility might not hold in the discrete token level, because of the existence of irreversible operations,* e.g. *the argmax operation discretizes probabilities to tokens and the CTC collapse process. But REDER still shows decent reconstruction capability in practice, as visually depicted in the experiment section.*

## 2.3 Training

Given a parallel corpus and a single model $\theta$, REDER can be jointly supervised by source-to-target and target-to-source translation for $f_{\theta}^{\rightarrow}$ and $f_{\theta}^{\leftarrow}$, respectively. Thus both translation directions can be achieved in one REDER model. We refer this to *bidirectional training*, which is opposite to *unidirectional training*, where each translation direction needs a separate model. Moreover, the reversibility of REDER enables appealing potentials to exploit consistency/agreement between forward and reverse directions. We introduce two auxiliary learning signals as follows.

**Layer-wise Forward-Backward Agreement** Since REDER is fully reversible, which consists of a series of computationally inverse of intermediate layers, an interesting question arises: given the desired output (*i.e.*, the target sentence), is it possible to derive the desired intermediate hidden representation by the backward target-to-source computation, and encourage the forward source-to-target intermediate hidden representations as close as possible to these "optimal" representations?

Given a source sentence $\boldsymbol{x}$, the inner representations of each layer in the forward direction are:

$$\overrightarrow{\mathbf{H}}_1 = \mathcal{F}_1(\boldsymbol{e}(\boldsymbol{x})), \ \overrightarrow{\mathbf{H}}_2 = \mathcal{F}_2(\overrightarrow{\mathbf{H}}_1) \ \ldots \ \overrightarrow{\mathbf{H}}_L = \mathcal{F}_L(\overrightarrow{\mathbf{H}}_{L-1}),$$

and given its corresponding target sequence $\boldsymbol{y}$ as the optimal desired output[4], the inner representations of each layer in the reverse direction are:

$$\overleftarrow{\mathbf{H}}_L = \mathcal{F}_L^{-1}(\boldsymbol{e}(y)), \ \overleftarrow{\mathbf{H}}_{L-1} = \mathcal{F}_{L-1}^{-1}(\overleftarrow{\mathbf{H}}_L) \ \ldots \ \overleftarrow{\mathbf{H}}_1 = \mathcal{F}_L^{-1}(\overleftarrow{\mathbf{H}}_2),$$

where $\overrightarrow{\mathbf{H}}_l$ and $\overleftarrow{\mathbf{H}}_l$ represent the representations of $l$-th layer in forward and reverse models, respectively. As we consider these reverse inner layer representations as "optimal", we try to minimize the cosine distance between the forward and backward corresponding inner layer representations:

$$\mathcal{L}_{\text{fba}}(\boldsymbol{x}|\boldsymbol{y};\theta) = \frac{1}{L} \sum_{l=1}^{L} 1 - cos(\overrightarrow{\mathbf{H}}_l, \mathtt{sg}(\overleftarrow{\mathbf{H}}_l)),$$

where $\mathtt{sg}(\cdot)$ denotes the stop-gradient operation.

**Cycle Consistency** The symmetry of a pair of seq2seq tasks enables the use of cycle consistency [He et al., 2016, Cheng et al., 2016a]. Given a source sentence $\boldsymbol{x}$, we first obtain the forward prediction, and then we use the REDER to reconstruct this prediction to the source language:

$$\bar{\boldsymbol{y}} = f_{\theta}^{\rightarrow}(\boldsymbol{x}), \quad \bar{\boldsymbol{x}} = f_{\theta}^{\leftarrow}(\bar{\boldsymbol{y}}).$$

Finally, we maximize the consistency or agreement between the original one $\boldsymbol{x}$ and reconstructed one $\bar{\boldsymbol{x}}$. Thus, the loss function reads

$$\mathcal{L}_{\text{cc}}(\boldsymbol{x};\theta) = \mathtt{distance}_{\text{cc}}(\boldsymbol{x}, f_{\theta}^{\leftarrow}(f_{\theta}^{\rightarrow}(\boldsymbol{x}))) = \mathtt{distance}_{\text{cc}}(\boldsymbol{x}, \bar{\boldsymbol{x}}).$$

We expect it can provide an auxiliary signal that a valid prediction should be loyal to reconstruct its source input. Here we use cross-entropy between the probabilistic prediction of the reverse model as distance to measure the consistency.

---

[4]for CTC-based model where the model predictions are the alignments, we instead extract the token sequence of the best alignment $\boldsymbol{a}^*$, predicted by the model, associated with the ground-truth $\boldsymbol{y}$ as the optimal desired output.

A potential danger of both of the above auxiliary objectives is learning a degenerate solution, where it would probably cheat this task by simply learning an identity mapping. We solve this problem by setting a two-stage training scheme for them, where we first train REDER without using any auxiliary losses until a predefined number of updates, and then activate the additional losses and continue training the model until convergence.

**Final Objective** Given a parallel dataset $\mathcal{D}_{\text{xy}} = \{(\boldsymbol{x}^{(n)}, \boldsymbol{y}^{(n)} | n = 1...N\}$ of i.i.d observations, the final objective of REDER is to minimize

$$
\begin{aligned}
\mathcal{L}(\theta; \mathcal{D}_{\text{xy}}) = \sum_{n=1}^{N} \Big( & -\log p_{\text{ctc}}(\boldsymbol{y}^{(n)} | \boldsymbol{x}^{(n)}; \theta) - \log p_{\text{ctc}}(\boldsymbol{x}^{(n)} | \boldsymbol{y}^{(n)}; \theta) \\
& + \underbrace{\lambda_{\text{fba}} \mathcal{L}_{\text{fba}}(\boldsymbol{x}^{(n)} | \boldsymbol{y}^{(n)}; \theta) + \lambda_{\text{fba}} \mathcal{L}_{\text{fba}}(\boldsymbol{y}^{(n)} | \boldsymbol{x}^{(n)}; \theta)}_{\textit{layer-wise forward-backward agreements}} \\
& + \underbrace{\lambda_{\text{cc}} \mathcal{L}_{\text{cc}}(\boldsymbol{x}^{(n)}; \theta) + \lambda_{\text{cc}} \mathcal{L}_{\text{cc}}(\boldsymbol{y}^{(n)}; \theta)}_{\textit{cycle consistencies}} \Big)
\end{aligned}
$$

where $\lambda_{\text{fba}}$ and $\lambda_{\text{cc}}$ are coefficients of the auxiliary losses.

## 3 Related Work

**Sequence-to-Sequence Models Exploiting Bidirectional Signals.** Several studies try to utilize bidirectionality as a constraint to improve sequence-to-sequence tasks such as machine translation [Cheng et al., 2016a,b]. Dual learning [He et al., 2016, Xia et al., 2017] leverages reinforcement learning to interact between two simplex translation models. Later, Xia et al. [2018] propose a partially model-level dual learning that shares some components of both models for forward and reverse tasks. Zheng et al. [2020] propose to model the two directional translation model with language models in a variational probabilistic framework. These approaches model two directional tasks by setting up two separate simplex models to consider the task bidirectionality. Different from them, REDER can unify a pair of directions within one duplex model and directly model the bidirectionality at a completely model level. Besides, other studies try to unify two directional tasks by multitask learning [Johnson et al., 2017, Chan et al., 2019] by sharing the same computational process of a single simplex model, which would inevitably result in parameter interference issue where two tasks compete for the limited model capacity. In contrast, REDER formulates both tasks in one model by simply exchanging input and output ends, each of which specializes in a language, thus bidirectional translation becomes a reversible process in which both directions do not need to compete for limited model capacity.

**Non-autoregressive Sequence Generation.** Non-autoregressive translation (NAT) models [Gu et al., 2018] aims to alleviate the decoding inefficiency of traditional autoregressive seq2seq models. Fully NAT models could generate sequence in parallel within only one shot but sacrifice performance [Ma et al., 2019, Shu et al., 2020, Bao et al., 2019, Wei et al., 2019, Qian et al., 2021, Gu and Kong, 2021, Huang et al., 2021]. Besides, semi-autoregressive models greatly improve the performance of NAT models, which perform iterative refinement of translations based on previous predictions [Lee et al., 2018, Ghazvininejad et al., 2019, Gu et al., 2019, Kasai et al., 2020, Ghazvininejad et al., 2020]. In this work, REDER takes the advantage of the probabilistic modeling of fully NAT models for resolving the designing challenge of computational homogeneity for both translation directions.

**Reversible Neural Architectures.** Various reversible neural networks have been proposed for different purposes. On one hand, reversible neural networks help model flexible probability distributions with tractable likelihoods [Dinh et al., 2014, 2017, Kingma et al., 2016], which define a mapping between a simple, known density and a complicated desired density. On the other hand, reversibility can also assist to develop memory-efficient algorithms. The most popular approach is the reversible residual network (RevNet, Gomez et al., 2017), which modifies the residual network and allows the activations at any given layer to be recovered from the activations at the following layer. Therefore layers can be reversed one by one as back-propagation proceeds from the output of the network to its input. Some follow-up work extends the idea of RevNet to RNNs [MacKay et al., 2018] and Transformer [Kitaev et al., 2020] in NLP. We borrow the idea of RevNet as the basic unit of our proposed REDER, however, for different purposes that we want to build a duplex seq2seq model to govern two directional tasks reversibly. In this line, van der Ouderaa and Worrall [2019] propose

a reversible GAN approach for image-to-image translation in computer vision, which to a certain extent shares the intuition with ours.

## 4 Experiments

We conduct extensive experiments on standard machine translation benchmarks to inspect REDER's performance on seq2seq tasks. We demonstrate that REDER achieves competitive results, if not better, compared to strong autoregressive (AT) and non-autoregressive (NAT) baselines. REDER is also the first approach that enables reversible machine translation in one unified model, where bidirectional training with paired translation directions surprisingly helps boost each of them with substantial gains.

### 4.1 Experimental Setup

**Datasets.** We evaluate our proposal on two standard translation benchmarks, *i.e.*, WMT14 English (EN) ↔ German (DE) (4.5M training pairs), and WMT16 English (EN) ↔ Romanian (RO) (610K training pairs). We apply the same prepossessing steps as mentioned in prior work (EN↔DE: Zhou et al., 2020, EN↔RO: Lee et al., 2018). BLEU [Papineni et al., 2002] is used to evaluate the translation performance for all models.

**Knowledge Distillation (KD).** Sequence-level knowledge distillation [Kim and Rush, 2016] is found to be crucial for training NAT models. Following previous NAT studies [Gu et al., 2018, Zhou et al., 2020], REDERs are trained on distilled data generated from pre-trained auto-regressive Transformer models. The beam size is set to 4 during generation.

**Beam Search Decoding and AT Reranking.** We implement two kinds of inference policies. The first one is parallel decoding that adopts tokens with the highest probability at each position. For CTC-based models, we also implement beam search to REDER with an efficient library of C++ implementation[5]. Similar to previous NAT literature, we further rerank the decoded candidates produced by beam search using autoreregressive models as the external scorer [Gu et al., 2018] and pick the best ones as final results.

**Implementation Details.** We design REDER based on the hyper-parameters of Transformer-*base* [Vaswani et al., 2017]. All models are implemented on `fairseq` [Ott et al., 2019]. REDER consists of 12 stacked layers. The number of head is 8, the model dimension is 512, and the inner dimension of FFN is 2048. For both AT and NAT models, we set the dropout rate $0.1$ for WMT14 EN↔DE and WMT16 EN↔RO. We adopt weight decay with a decay rate $0.01$ and label smoothing with $\epsilon = 0.1$. By default, we upsample the source input by a factor of 2 for CTC-based models. We set $\lambda_{\text{fba}}$ and $\lambda_{\text{cc}}$ to 0.1 for all experiments. All models are trained for 300K updates using Nvidia V100 GPUs with a batch size of approximately 64K tokens. Following prior studies [Vaswani et al., 2017], we compute tokenized case-sensitive BLEU. We measure the validation BLEU scores for every 2,000 updates, and average the best 5 checkpoints to obtain the final model. Similar to previous NAT studies, we also measure the GPU latency by running the model with one sentence per batch on WMT14 EN-DE test set on a single GPU and give speedup comparing over our AT baselines.[6]

### 4.2 Main Results

As shown in Table 1, we compare REDER with AT and NAT approaches with and without multitask learning, as well as existing approaches that also leverage bidirectional learning signals.

**REDER achieves competitive results compared with strong NAT baselines.** We show that a unified REDER trained on the same parallel data can simultaneously work in two directions, which has a comparable capability as strong NAT models such as GLAT [Qian et al., 2021] (row 14 *vs*. row 8). With the help of beam search and re-ranking, the performance of REDER can be further boosted to be comparable with the state-of-the-art NAT method GLAT+CTC [Gu and Kong, 2021] (row 14 *vs*. row 9 & 10). Gu and Kong [2021] explore the best technique combination for NAT, whose tricks can also supplement to enhance REDER. We leave this for exploration.

---

[5]`https://github.com/parlance/ctcdecode`

[6]Note that as this paper's goal is not for decoding efficiency, all models are not optimized for latency using advanced techniques, and the autoregressive baselines are hence weak in terms of latency.

Table 1: Comparisons between our models and existing models. All NAT models are trained with KD. "↔": whether to allow bidirectional translation. "MTL": multitask learning. "BT": back-translation. The speedup is measured with batch size of 1. Notice that speedups from previous papers are generally not fully comparable due to inconsistent hardware and baselines and hence only for reference. All our implemented CTC-based NAT models (row 10 ∼ row 14) employ beam search decoding with beam size of 20, whereas NAT models from previous literature (row 4 ∼ row 9) employ greedy decoding.

| | Systems | ↔ | $|\theta|$ | Speed | WMT14 | | WMT16 | |
| | | | | | EN-DE | DE-EN | EN-RO | RO-EN |
|---|---|---|---|---|---|---|---|---|
| AT | ① Transformer-*base* (KD teacher) | ✗ | 62M× 2 | 1.0× | 27.60 | 31.50 | 33.85 | 33.70 |
| | ②  w/ *MTL* | ✓ | 62M | 1.0× | 27.06 | 30.96 | - | - |
| | ③  w/ *BT* | ✗ | 62M×2 | 1.0× | 27.82 | 31.91 | - | - |
| NAT | ④ vanilla NAT [Gu et al., 2018] | ✗ | 62M×2 | 15.6× | 17.69 | 21.47 | 27.29 | 29.06 |
| | ⑤ CTC w/o KD [Libovický and Helcl, 2018] | ✗ | 58M×2 | - | 16.56 | 18.64 | 19.54 | 24.67 |
| | ⑥ *CTC* [Saharia et al., 2020] | ✗ | 58M×2 | 18.6× | 25.70 | 28.10 | 32.20 | 31.60 |
| | ⑦ *Imputer* [Saharia et al., 2020] | ✗ | 58M×2 | 18.6× | 25.80 | 28.40 | 32.30 | 31.70 |
| | ⑧ *GLAT* [Qian et al., 2021] | ✗ | 62M×2 | 15.3× | 26.55 | 31.02 | 32.87 | 33.51 |
| | ⑨ *GLAT+CTC* [Gu and Kong, 2021] | ✗ | 62M×2 | 16.8× | 27.20 | **31.39** | **33.71** | **34.16** |
| | *our re-implementations of* Gu and Kong [2021]: | | | | | | | |
| | ⑩ *GLAT+CTC* | ✗ | 62M×2 | 16.2× | 26.79 | 30.45 | - | - |
| | ⑪  w/ *MTL* | ✓ | 62M | 16.2× | 25.50 | 29.49 | - | - |
| REDER | ⑫ *simplex* REDER | ✗ | 58M×2 | 5.5× | 26.20 | 30.02 | 32.67 | 32.98 |
| | ⑬  w/ *MTL* | ✓ | 58M | 5.5× | 25.58 | 29.12 | - | - |
| | ⑭ *duplex* REDER (final model) | ✓ | 58M | 5.5× | **27.50** | 31.25 | 33.60 | 34.03 |
| previous studies | Reformer [Kitaev et al., 2020] | ✗ | 62M×2 | - | 27.60 | - | - | - |
| | Model-level DL *big* [Xia et al., 2018] | ✗ | 210M×2 | - | 28.90 | 31.90 | - | - |
| | KERMIT [Chan et al., 2019] | ✗ | 124M | - | 25.60 | 27.40 | - | - |
| | KERMIT + mono [Chan et al., 2019] | ✗ | 124M | - | 28.10 | 28.60 | - | - |
| | MGNMT [Zheng et al., 2020] | ✓ | 195M | - | 27.70 | 31.40 | 32.70 | 33.90 |

**Duplex learning has more potential than multitask learning and back-translation.** Given the same parallel corpus as training data, multitask learning (MTL) yields considerable performance degradation of either AT (row 1 *vs*. row 2). By re-implementing GLAT+CTC [Gu and Kong, 2021] (row 10 & 11) as the strong NAT competitor for more convincing comparison. We observe that MTL would hurt more severely for NAT models, such as GLAT+CTC models (row 11 *vs*. row 10), and REDER as well (row 13 *vs*. row 12). These results verify our concern of multitask-learned models regarding parameter interference. Meanwhile, when no external monolingual resources are available, back-translation (BT) only adds mild points from training data. In contrast, duplex learning allows REDER to gain more benefits, becoming a better alternative and a parameter-efficient choice to exploit more potentials from the provided parallel data. Plus, duplex learning is orthogonal to BT, which could further improve REDER with monolingual data.

**REDER performs on par with autoregressive Transformer.** Despite the challenge in regards to non-autoregressive modeling and non-encoder-decoder design, REDER closely approaches the simplex AT models (row 14 *vs*. row 1), while REDER can translate both directions in one model. Besides, REDER even surpasses the multitask-learned bidirectional autoregressive Transformer model (row 14 *vs*. row 2). Additionally, REDER enjoys faster decoding spend than the baseline autoregressive models. This evidence shows the advantage and practical value of reversible machine translation as a more decent solution for parameter-efficient bidirectional translation systems.

**Comparison with existing approaches.** Reformer [Kitaev et al., 2020] also employs RevNet to make parts of the Transformer model to reduce the memory consumption for training, while REDER is fully reversible with a different motivation of maximizing the use of bidirectional signals. Existing simplex approaches exploiting bidirectional signals require two separate simplex models for both directions [Xia et al., 2018, Zheng et al., 2020]. REDER, in contrast, only needs one unified duplex model and coherently models two directions. Alternatively, Chan et al. [2019] use a single simplex network to achieve bidirectional translation via multitask learning, needing to split limited capacity for both directions, which underperform REDER on parallel settings.

**Training cost.** We train REDER on WMT14 EN↔DE using 8 32GB V100 GPUs for 432 hours (54 hour per GPU) and obtained a bidirectional translation model. For modeling both directions, a standard NAT model needs 640 GPU hours (320×2) in total, whilst the autoregressive Transformer needs 400 GPU hours (200×2) using the same computational resources. Therefore, the training costs of these methods are comparable.

### 4.3 Ablation Study of Model Design

REDER is developed on the top of various components in terms of data (knowledge distillation), learning objective (CTC), architecture (revnet, relative attention), and auxiliary losses endowed by reversibility of REDER. We analyze their effects through various combinations in Table 2. We first consider training REDER for a single direction to seek the best practice to run REDER for sequence-to-sequence tasks. KD and CTC are essential to training REDER, as suggested by previous NAT studies [Saharia et al., 2020, Gu and Kong, 2021]. Meanwhile, we notice the benefit of relative self-attention. We therefore use these three techniques by default for all of the proposed models. As for the duplex variants of models that learn both directions simultaneously, they can further improve the translation accuracy by substantial margins. These results verify our motivation that the paired translation directions could be better learned in a unified reversible model. Reversibility enables us to utilize layer-wise forward-backward agreement and cycle consistency, which are also shown to boost improvement considerably.

Table 2: Ablation on WMT14 EN→DE test set with different combinations of techniques. R-SAN denotes relative self attention. For fair comparison, all CTC and non-CTC variants do not use beam search decoding.

| KD | CTC | revnet | R-SAN | $\mathcal{L}_{\text{fba}}$ | $\mathcal{L}_{\text{cc}}$ | BLEU |
|----|-----|--------|-------|------|------|-------|
|    |     |        |       |      |      | 11.40 |
| ✓  |     |        |       |      |      | 19.50 |
|    | ✓   |        |       |      |      | 16.90 |
| ✓  | ✓   |        |       |      |      | 25.01 |
| ✓  | ✓   |        | ✓     |      |      | 25.55 |
| ✓  | ✓   |        | ✓     | ✓    |      | 25.90 |
| ✓  | ✓   | ✓      | ✓     |      |      | 26.20 |
| ✓  | ✓   | ✓      | ✓     | ✓    |      | 26.65 |
| ✓  | ✓   | ✓      | ✓     |      | ✓    | 26.70 |
| ✓  | ✓   | ✓      | ✓     | ✓    | ✓    | **26.89** |

### 4.4 Decoding: Effect of Beam search and Re-ranking

The performance of REDERs can be further boosted with additional (beam-search or re-ranking) techniques. For CTC beam search, we use the teacher model (AT *base*) to re-rank the translation candidates obtained by the beam search to determine the one with the best quality. As shown in Table 3, a larger beam size results in a smaller BP for AT models, meaning it produces shorter translations [Stahlberg and Byrne, 2019, Eikema and Aziz, 2020]. CTC beam search helps REDER produce longer outputs (larger BPs) but only endows a little improvement. With beam search and AT reranking, REDER can generate more decent translations. These results imply that we need to find a better way to train REDER (and probably the NAT family) if we do not want to involve an extra AT for such a somewhat inconvenient re-ranking.

Table 3: Comparisons regarding decoding methods for REDER on WMT14 EN↔DE. The brevity penalty (BP) given by BLEU indicates the adequacy of translation: the lower the BP, the more inadequate the translation.

| Systems | EN-DE | DE-EN | BP | Speed |
|---------|-------|-------|-----|-------|
| Transformer (AT, teacher) | 27.20 | 31.00 | 0.980 | 1.0 × |
|   + *beam=5* | 27.60 | 31.50 | 0.998 | - |
|   + *beam=20* | 27.65 | 31.12 | 0.954 | - |
| REDER w/ greedy decoding | 26.89 | 30.90 | 0.935 | 19.8 × |
|   + *beam=20* | 26.90 | 30.75 | 0.995 | 6.8 × |
|   + *beam=20* + AT reranking | 27.50 | 31.25 | 1.000 | 5.5 × |
|   + *beam=100* | 26.92 | 30.95 | 0.982 | 2.1 × |
|   + *beam=100* + AT reranking | 27.59 | 31.45 | 1.000 | 1.2 × |

### 4.5 Training: Impact of Knowledge Distillation

Like other NAT approaches, we find that REDER heavily relies on knowledge distillation. We report the performance of models trained on raw data and distilled data generated from AT models in Table 4. As we can see, without KD, the accuracy of REDER significantly drops. We then aim to explore the most proper way to integrate KD data. We observe that if we only use KD data of one direction (only target-side data are distilled, *e.g.*, German sentences in EN-DE), it only benefits a single direction. These imply that we need to provide KD data of both directions to train REDER. Furthermore, we notice that if we mix the KD data of both directions by concatenating them directly, it somehow improves results compared to the strategy only using single-direction KD data. Finally, we find the best practice is to separately feed KD data in accordance to directions, *i.e.*, feeding EN-DE KD data when training the EN-DE direction and providing DE-EN KD data when training the reverse direction (*i.e.*, DE-EN).

Table 4: Performance regarding KD on WMT14 EN↔DE. #data means the amount of data points for each direction.

| Systems | EN-DE | DE-EN | #data |
|---------|-------|-------|-------|
| all raw | 17.85 | 19.68 | *N/N* |
| EN→DE KD (only DE distilled) | 25.49 | 26.57 | *N/N* |
| DE→EN KD (only EN distilled) | 23.04 | 28.82 | *N/N* |
| mixture KD | 26.50 | 29.65 | *2N/2N* |
| separate KD (final model) | 27.50 | 31.25 | *N/N* |

**Discussion.** Like other NAT approaches, the proposed REDER resorts to, and unfortunately heavily relies on, KD data for training. Requiring KD does hinder the generalization of NAT models including REDER to other applications especially multilingual scenarios. We notice that recent studies could have the potentials for the removal of KD for NAT models, through introducing latent-variable models [Gu and Kong, 2021, Bao et al., 2019] or sampling/denoising-based augmented training objectives/strategies [Qian et al., 2021] As KD-dependence is a common issue for all NAT approaches, we believe future breakthroughs would resolve this. Please note that the aim of this paper is to make the idea of duplex sequence-to-sequence learning and its implementation of REDER realizable, at least in the scenario with KD data. Eliminating the need for KD is orthogonal to the purpose of this paper, however, is very valuable for further exploration.

### 4.6 Analysis of Reversibility

We examine the reversibility of REDER by both quantitative and qualitative analysis. We first measure the BLEU score between source sentences $x$ and the associated reconstructions, *i.e.* BLEU$(x, f_\theta^\leftarrow(f_\theta^\rightarrow(x)))$, on the development sets of WMT14 EN-DE, which gives a score of 66.0. Besides, we also show an example regarding its forward prediction and reconstruction in Figure 3. Given a sentence in the source language (EN), we first use the forward mapping of REDER to obtain a prediction in the target language

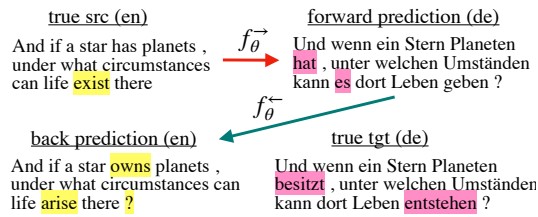

Figure 3: Case study of reversibility.

(DE), and then translate it back to the source language using the reverse model. As shown in Figure 3, REDER can reconstruct the input from the output with mild differences to some extent. These results demonstrate that REDER meets the definition of reversibility empirically to a certain extent.

### 4.7 Experiments on Distant Languages

To examine whether REDER can generalize to distant languages, we conducted experiments on WMT20 English-Japanese (EN↔JA), where the training data is much larger and two languages are linguistically distinct with almost no vocabulary overlap. As shown in Table 5, REDER can achieve very close results compared with AT in such a large-scale scenario with distant languages, showing that reversible machine translation could make more potentials of parallel data.

Table 5: Results of WMT20 EN↔JA (∼16M). Here REDER uses beam search with b=20 and AT-reranking.

| Systems | EN-JA | JA-EN | ↔ |
|---|---|---|---|
| AT *big* | 20.3 | 21.5 | ✗ |
| REDER *big* | 20.0 | 20.7 | ✓ |

## 5  Conclusion and Future Work

In this paper, we propose REDER, the **RE**versible **D**uplex Transform**ER** for sequence-to-sequence problem and apply it to machine translation that for the first time shows the feasibility of a reversible machine translation system. REDER is a fully reversible model that can transform one sequence to the other one forth and back, by reading and generating through its two ends. We verify our motivation and the effectiveness of REDER on several widely-used NMT benchmarks, where REDER shows appealing performance over strong baselines.

As for promising future directions, REDER can be applied to monolingual, multilingual and zero-shot settings, thanks to the fact that each "end" of REDER specializes in a language. For instance, given trained REDERs $\mathcal{M}_{En \leftrightarrow De}$ and $\mathcal{M}_{En \leftrightarrow Ja}$, we combine last half layers (the DE end) of $\mathcal{M}_{En \leftrightarrow De}$ and the JA end of $\mathcal{M}_{En \leftrightarrow Ja}$ to obtain a zero-shot $\mathcal{M}_{De \leftrightarrow Ja}$, translating between German and Japanese. Likewise, the composition of an English end and its reverse results in $\mathcal{M}_{En \leftrightarrow En}$, which can learn from monolingual data like an autoencoder. This compositional fashion resembles LEGO, which manipulates only a linear number of language ends. Therefore, while adding a new language to a multilingual REDER system (in a form of composition of ends of involved languages), we would probably not need to retrain the whole system as we do for a current multilingual NMT system, which reduces the difficulty and cost to train, deploy and maintain a large scale multilingual NMT system.

## Acknowledgements

We would like to thank the anonymous reviewers for their insightful comments. Hao Zhou is the corresponding author. This work was supported by National Science Foundation of China (No. 61772261, 6217020152), National Key R&D Program of China (No. 2019QY1806).

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
