# OpenReview forum: "Duplex Sequence-to-Sequence Learning for Reversible Machine Translation"
_NeurIPS.cc/2021/Conference — NeurIPS 2021 Poster_

### Official Review · Reviewer_Bs9X · 2021-07-12

**Rating:** 6
**Confidence:** 5

**Summary:**

The paper proposes a novel machine translation model that can be simultaneously applied to bidirectional translations. The model, named REDER, is symmetric at both input and output ends and consists of several identical reversiable duplex Transformer layers. REDER adopts the design of revnet for reversibility and reverses the first L/2 layers for homogeneity, which is a fully non-autoregressive NMT model. Extensive experiments on WMT14 En$\leftrightarrow$De and WMT16 En$\leftrightarrow$Ro datasets show that the proposed model consistently outperforms (non-)autoregressive baselines with improving decoding speed.

**Limitations And Societal Impact:**

There is no explicit presentation of limitations or potential negative societal impact in this paper.

**Main Review:**

Strengths

This work to clearly identify shortcomings in the current literature. Experimental results seem to be solid. The identification of REDER as a possible approach to realize reversible machine translation.

Questions

* There is a definition of duplex sequence-to-sequence neural netwrok in line 57, and the proposed REDER is also optimized by the cycle consistency. However, there is no experiments to verify the resulted REDER model is satisfied with this pre-definition. Can you provide some quantitative results on the development sets, such as the BLEU or word prediction accuracy of the reconstructed outputs, e.g. $f_{\theta}^{\leftarrow}(f_{\theta}^{\rightarrow}(x))$ or $f_{\theta}^{\rightarrow}(f_{\theta}^{\leftarrow}(y))$.
* Existing multilingual machine translation models can also realize reversible machine translation that uses language embedding to avoid parameter interference problem. Can you provide more comparisons between REDER and multilingual NMT models (only involves the source and target languages.)
* A related work is missing, i.e. supervised dual learning [Xia et al., 2017: ICML].
* Why duplex neural networks can avoids the parameter interference problem? (Statement in line 51-52) It is not very clear for me.
* Writing could have been clearer. What the superscript (1) and (2) in Figure 2 mean? Why $y^{(1)}$ is dependent on $x^{(2)}$ and $y^{(2)}$ is dependent on $x^{(1)}$.
* I would to see the paragraph of $\textit{Modeling variable-length input and output}$ to be put behind $\textit{The Architecture of REDER}$, which is more clearer for me.

Minor comments

* In Definition 1, a closing bracket is missing in the last sentence, i.e. $f_{\theta}^{\leftarrow}(f_{\theta}^{\rightarrow}(x))=x$.
* In line 74, change "multitask learning with" to "With multitask learning".


**Time Spent Reviewing:**

3.5

---

> ### Author Response · Authors · 2021-08-10
> **Response to Reviewer Bs9X**
>
> Thanks very much for your valuable comments!
>
> &nbsp;
>
> **Q1:** There is a definition of duplex sequence-to-sequence neural netwrok in line 57, and the proposed REDER is also optimized by the cycle consistency. However, there is no experiments to verify the resulted REDER model is satisfied with this pre-definition. Can you provide some quantitative results on the development sets, such as the BLEU or word prediction accuracy of the reconstructed outputs, e.g., $(f^{\leftarrow}_\theta(f^{\rightarrow}_\theta(x), x)$ or $(f^{\rightarrow}_\theta(f^{\leftarrow}_\theta(y), y)$.
>
> **A1:**  Thanks for your suggestions. We did a quick quantitative analysis as you suggested. The BLEU score between x and its reconstruction, i.e. $\text{BLEU}(x, f_\theta^{\leftarrow}f_\theta^{\rightarrow}(x))$, on the development sets of WMT14 En-De is **66.0**. Besides, as the case shown in Figure 2 in the appendix, REDER could to a certain extent recover the prediction to the input sequence. These results demonstrate that REDER meets the definition empirically to a certain extent.
>
> Besides, sorry for any confusion this may have caused. REDER is exactly reversible between the input embedding and the output embedding (i.e. the hidden representations before softmax); and it is not exactly reversible between the source sentence and target sentence due to existing irreversible operations (e.g., CTC collapse process).
>
> To be clearer, considering $x$ and $y$ as a pair of source and target sentences, $emb(x^{upsampled})$ is the input embedding of the upsampled $x$ and $H^{L}$ is the output hidden representations of the network that correspond to the predicted CTC alignment $a$. REDER allows reversibility between $emb(x^{upsampled})$ and $H^{L}$ by its architecture design. However, we cannot assure the exact reversibility in the discrete token level beween $x$ and $y$, because of the irreversible argmax operation discretizing probabilities to tokens, as well as the irreversible CTC collapse.
>
> &nbsp;
>
>
> **Q2:**  Existing multilingual machine translation models can also realize reversible machine translation that uses language embedding to avoid parameter interference problem. Can you provide more comparisons between REDER and multilingual NMT models (only involves the source and target languages.)
>
> **A2:** Actually, the multitask baseline (MTL) we used has adopted the language embedding you mentioned, which was indeed a multilingual NMT model with only two languages involved.
>
> We will provide more discussion between REDER and multilingual NMT regarding parameter interference problem in the next question.
>
> &nbsp;
>
>
> **Q3:** Why duplex neural networks can avoids the parameter interference problem? (Statement in line 51-52) It is not very clear for me.
>
> **A3:** Generally, REDER could address the parameter interference because its specific ends process specific languages correspondingly, instead of using one end to process two languages as in multilingual NMT. More specifically,
>
> - In a multilingual NMT (MNMT) model, the shared encoder and decoder are required to understand and generate two different languages (e.g., Japanese and English) simultaneously. Translation from Japenese to English and English to Japenese are quite different since they are linguistically distant. Despite adding language embeddings to differentiate languages, the rest of the model parameters and computational process are still shared for two translation directions. For example, imagine that the decoder of an MNMT is required to learn to speak English with a word order of SVO, while also learn to speak Japanese with a discrepant word order of SOV. Therefore, the two translation directions will compete for the limited model capacity (the parameter interference issue), leading to worse performance.
>
> - On the contrary, REDER formulates both tasks in one model by simply exchanging input and output ends, each of which specializes in a language. Intuitively, In either English-to-Japenese or Japanese-to-English translation, the English end and Japanese end only need to learn to "care about their own business" as reading and speaking a specific language (i.e., the English end only needs to read and speak English following the SVO word order, and so does the Japanese end), which has been modeled as a unified and reversible process by our design, and does not need to compete but better utilize the model capacity.
>
> Although we have discussed the differences in the Related Work section, we will make this clearer in the next version.
>
> &nbsp;
>
>
> **Q4:** What the superscript (1) and (2) in Figure 2 mean? Why is $y^{(1)}$ dependent on $x^{(2)}$ and is $y^{(2)}$ dependent on $x^{(1)}$?
>
> **A4:** The input and output of a revnet (Gomez et al, 2017) block are split by 2 halves, i.e., $(x^{(1)}, x^{(2)})$ and $(y^{(1)}, y^{(2)})$, which are what the superscripts mean in Figure 2. Such a design is proposed for the purpose of achieving reversibility.
>
> Formally, the forward compute of a revnet block is executed as following:
>
> $$y^{(1)} = x^{(1)} + F(x^{(2)}) $$
> $$y^{(2)} = x^{(2)} + G(y^{(1)})$$
>
> In such a way one can obtain the input $(x^{(1)}, x^{(2)})$ reversely if given the output $(y^{(1)}, y^{(2)})$ by algebra:
>
> $$x^{(2)} = y^{(2)} - G(y^{(1)})$$
> $$x^{(1)}= y^{(1)} - F(x^{(2)})$$
>
> As a result, a revnet block $R: (x^{(1)}, x^{(2)}) \rightarrow (y^{(1)}, y^{(2)})$ is a bijection. Similar ideas are widely adopted in other reversible architectures such as flow-based models (e.g., RealNVP (Dinh et al, 2017)). \
> Compared to an ordinary irreversible neural block counterpart $F \circ G: x \rightarrow y$, where $dim(x^{(1)}) = dim(x^{(2)}) = dim(x)$ and $dim(y^{(1)}) = dim(y^{(2)}) = dim(y)$, the dimmension of the inner hidden representations is unchanged thus the computational overhead remains comparable. In REDER, particularly, the input embedding of the first layer $(x^{(1)}, x^{(2)}) = (x, x)$ where $(x, x)$ is two copies of embedding of $x$.
>
>
> &nbsp;
>
>
> **Q5:** About citation of [Xia et al, ICML17] and other presentation suggestions
>
> **A5:** Actually, we have cited the mentioned reference in the Related Work section in the submission. And we thanks for pointing out all the writing mistakes and presentation suggestions, we will fix them and improve the presentation in the next revision.
>
> ---
> **References:**
> - Gomez et al, 2017. The Reversible Residual Network: Backpropagation Without Storing Activations. In NeurIPS
> - Dinh et al, 2017. Density estimation using Real NVP. In ICLR

---

### Official Review · Reviewer_19ey · 2021-07-17

**Rating:** 6
**Confidence:** 4

**Summary:**

This paper proposes a duplex reversible network based on Transformer and revnet for neural machine translation. With the proposed approach, one can use one single model to translate between a language pair from both directions. The model uses CTC loss and allows non-autoregressive decoding for faster inference time. Experiments are well executed on standard NAT benchmark datasets with close performance to that of the standard Transformer model.

**Contributions**: the proposed model is new in machine translation from the best of my knowledge and achieves good performance compared to non-autoregressive models and multi-task learning. The proposed method saves parameters in terms of translation from both directions and improves inference speed.

**Ethical Concerns:**

N/A.

**Limitations And Societal Impact:**

N/A.

**Main Review:**

**Originality**: the proposed work resembles the coupling network in the literature of flow-based models; authors adapt the revnet to Transformer models by designing a fully symmetric architecture that accepts inputs and emits outputs from both ends.

**Clarity**: the paper is generally well written but contains many typos. I'll list a few here: L47 -> interference, L50 -> language, L62 -> (), L74 -> the sentence of multitask ..., L77 -> adopts, L79 -> bonus also influent sentence; please double-check the writing.

**Weakness and Questions**:
1. Authors mentioned one advantage of the proposed method in the introduction which is it works well for linguistically distant language pairs; I did find the results for ja-en in the appendix, but I think these are pretty interesting and important results and should be moved to the main paper.

2. It seems that the training cost of the duplex transformer could higher than standard training. Could you report approximately the training time for the proposed model versus. the Transformer model? Besides, **it heavily relies on distillation data**, which makes me worry about its broader application to other MT scenarios, e.g. multilingual MT.

3. The discussed future work seems to be interesting. Adding one such experiment for zero-shot or multilingual MT would make the paper stronger. There could be plenty of interesting concerns when applying this model to multilingual MT. I'd like to see if the duplex transformer would outperform standard multilingual training.

4. The other related approach which could potentially reduce the parameters is parameter sharing and separation for different language pairs, where you could share many parameters for both language pairs and have individual parameters (like adapters) for individual language pair to alleviate the parameter interference problem.

**Time Spent Reviewing:**

3

---

> ### Author Response · Authors · 2021-08-10
> **Response to Reviewer 19ey**
>
> Thanks very much for your valuable comments! We have tried to address all your concerns as following, especially adding results for zero-shot and multilingual MT setting. Please have a check and any further feedbacks are welcome!
>
> &nbsp;
>
>
> **Q1:** Results of distant language pairs of ja-en should be moved to the main paper
>
> **A1:** Thanks for the suggestions and we will revise as suggested.
>
> &nbsp;
>
>
> **Q2:** Concerns about the training cost of REDER
>
> **A2:** We trained REDER on WMT14 EN<->DE using 8 32GB V100 GPUs for 54 GPU hours and obtained a bidirectional translation model.  For modeling both directions, a standard NAT model needed 80 GPU hours (40x2) while the autoregressive Transformer needed 50 GPU hours (25x2) using the same computational resources. Therefore, the training time of these methods are comparable.
>
> &nbsp;
>
>
> **Q3:** Adding experiments for zero-shot or multilingual MT settings would make the paper stronger and would be interesting.
>
> **A3:** Thanks for your kind suggestions. We've conducted experiments on multilingual settings and observed that REDER can obtain promising results.
>
> **About experimental settings.**
> - The experiments were performed on German↔English (de↔en) and French↔English (fr↔en) datasets from IWSLT'17 multilingual benchmark, with Transformer-base as model hyperparameters. The bilingual and multilingual baselines are autoregressive models,  The multilingual REDER used KD data provided by bilingual Transformers.
> - Besides, We also conducted experiments on zero-shot directions, i.e., German↔French (de↔fr), where the testsets were constructed by extracting the German and French translation pairs from de↔en and fr↔en testsets if they share the same English translations.
>
> The results are summarized in the following table.
>
> | Models                   |         |       |   |         |       |   | |  zero-shot     |
> |--------------------------|--------:|------:|--:|--------:|------:|--:|----------:|-----:|
> |                          |  |   de ↔ en    |   |  |    fr ↔ en   |   |   |   de ↔ fr    |
> |                          |       ← |     → |   |       ← |     → |   |         ← |    → |
> | bilingual                |   29.52 | 36.61 |   |    32.1 | 31.67 |   |         - |    - |
> | multilingual Transformer |   28.36 | 35.05 |   |   30.44 | 30.27 |   |       3.2 |  4.6 |
> | mREDER                   |   29.12 |  36.7 |   |    31.6 | 31.44 |   |      15.4 | 21.7 |
>
>
> - **About multilingual results.** As we can see, when introducing more languages, multilingual REDER (mREDER) surpasses the multilingual autoregressive Transformer with a large margin and achieves very comparable results with the autoregressive bilingual baseline. These results reveal that REDER should be a promising competitor for multilingual translation.
> - **About zero-shot results.** We find that the vanilla multilingual NMT model suffers from a severe "off-target" problem on zero-shot directions, where the translations are generated into English other than the desired target language. This problem is also often observed by previous studies (Zhang et al, 2020; Lin et al 2021), while this result is consistent with the latest results from Table 3 of Lin et al., (2021).\
> On the contrary, mREDER could alleviate this problem since each of its language ends focuses on learning to read and generate a specific language, such that an unseen translation direction can be achieved by combining the source language end and the reverse of the target language end. This result shows that multilingual REDER has the potential to become a novel way for multilingual and zero-shot MT by manipulating its language ends, which is very interesting and worth further exploration.
>
> We will keep working on this and update more results in the future.
>
>
> &nbsp;
>
>
> **Q4:** Related approach which could potentially reduce parameter interference by parameter sharing and separation for different language pairs, e.g., having individual parameters like adapters for individual language pair.
>
> **A4:** Thanks for your insightful comments. This line of solution seems to be interesting and useful for alleviating the parameter interference issue, while REDER is obviously not the only solution for the parameter interference. We will add related discussions in the revision.
>
> &nbsp;
>
>
> **Q5**: Heavy reliance on KD data
>
> **A5**: Thanks for your questions. Please refer to the general response.
>
> &nbsp;
>
>
> **Q6:**: About presentation suggestions
>
> **A6:** Thanks for pointing out all the writing mistakes and presentation suggestions, we wil fix them and improve the presentation in the next revision.
>
> ---
> **References:**
> - Zhang et al. 2020. Improving massively multilingual neural machine translation and zero-shot translation. In ACL.
> - Lin et al. 2021. Learning Language-Specific Sub-network for Multilingual Machine Translation. In ACL

---

> > ### Comment · Reviewer_19ey · 2021-08-29
> > **Thanks for your reponse!**
> >
> > Thanks for the clarifications and additional experiments! I am satisfied with the response, hence increasing my score by 1.

---

> > > ### Author Response · Authors · 2021-08-30
> > > **Thank you!**
> > >
> > > We again highly appreciate your insightful and helpful comments, and thanks for increasing the score!

---

### Official Review · Reviewer_LGLR · 2021-07-19

**Rating:** 7
**Confidence:** 4

**Summary:**

This paper proposes an architecture or a reversible NMT system. In contrast to an encoder-decoder network, the two components of the network for each language are identical. When a language is being analyzed or generated depends on the direction of data flow in the network. To enable this, the authors propose reversible components in each layer of the network (inspired from revnet). The model is trained with CTC loss along with auxiliary losses for cyclic consistency and layerwise agreement. The paper is an interesting proposal for a reversible NMT system, and it brings together work on revnet, CTC loss, NAT in a simple, elegant framework to enable reversible NMT.

**Limitations And Societal Impact:**

Nothing to comment

**Main Review:**

While the architecture is novel and interesting, it is not clear what implication does a reversible network have for sequence to sequence learning and MT.
- The authors mention that reversible NMT can be useful for using supervised signals in both directions to improve translation quality. However, the results do not seem to substantiate this. The gains using REDER are none to minor over an MTL baseline. Further, the use of cyclic consistency loss for the joint training and the alignment losses also do not seem to help to any meaningful extent.
- The other benefit pointed out by the authors is the decoding speedup on account of the non-autogressive nature of the solution. The speedup is much lower compared to the other NAT approaches, while the translation quality is comparable. REDER uses similar techniques as CTC-based NAT approaches - why is it then about 3x slower than these models? Can the authors elaborate on this discrepancy?
- Like other NAT approaches, the model still relies on KD data to be competitive with auto-regressive models. This seems to be one sore spot in an otherwise elegant model.

Overall, I think the paper puts together many ideas and proposes an interesting idea of reversible NMT and an elegant solution. The results do not such much benefit in terms of benefits from bilingual signals or decoding speedup. However, this is an interesting proposal and framework to further explore.

**Time Spent Reviewing:**

2

---

> ### Author Response · Authors · 2021-08-10
> **Response to Reviewer LGLR**
>
> Thanks very much for your insightful comments!
>
> &nbsp;
>
>
> **Q1:** The authors mention that reversible NMT can be useful for using supervised signals in both directions to improve translation quality.
> However, the results do not seem to substantiate this. The gains using REDER are none to minor over an MTL baseline. Further, the use of cyclic consistency loss for the joint training and the alignment losses also do not seem to help to any meaningful extent.
>
> **A1:** We try to answer these questions as follows:
> - **About the comparison with non-autoregressive MTL baselines.** REDER achieves an absolute 2.0 BLEU gain compared to our implemented GLAT+CTC model w/ MTL (27.5 v.s 25.50, row 14 vs row 11).
> - **About the comparison with autoregressive Transformer+MTL baselines**. REDER can also obtain a 0.44 improvement compared to AT+MTL (27.5 vs 27.06, row 14 vs row 2). \
> These results indicate that REDER can make better use of bidirectional signals than AT-MTL and NAT-MTL approaches.
> - **About the impact of the auxiliary losses.** From the last four rows in Table 2, cycle consistency and layer-wise agreement losses bring 0.5 and 0.45 improvement in BLEU, respectively. Adding both leads to a total of 0.69 improvements, showing that they contribute considerably to the performance of REDER.
>
> &nbsp;
>
> **Q2:**  Why is it then about 3x slower than other CTC-based models? Can the authors elaborate on this discrepancy?
>
> **A2:** Sorry for the confusion.
> - **About the discrepancy of speed**. We reported REDER models with beam search in Table 1, which reduces the decoding speed. The cited CTC-based approaches in Table 1 (e.g, GLAT+CTC, Imputer) used greedy decoding, thus having a faster speed.
> - **About the comparison on greedy decoding**. For As shown in Table 3, when using greedy decoding, REDER could also achieve a similar speedup (19.8x) with a BLEU score of 26.89. Meanwhile, in Gu and Kong (2020), their CTC-only NAT model obtains a BLEU score of 26.51. And their GLAT+CTC model gains a BLEU score of 27.20 if incorporating glancing training technique (Qian et al, 2020). \
> These reveal that REDER is also competitive when using greedy decoding and not using other advanced training techniques. Besides, we believe that the best practice of training NAT models by Gu and Kong (2020) could also supplement improving REDER. We will leave this for further study.
>
> &nbsp;
>
>
> **Q3:** NAT models still rely on KD data
>
> **A3:** Thanks for your questions. Please refer to the general response.
>
> &nbsp;
>
> **Q4:** About presentation suggestions
>
> **A4:** Thanks for pointing out all the writing mistakes and presentation suggestions, we wil fix them and improve the presentation in the next revision.
>
> ---
>  References:
> - Gu and Kong, 2021. Fully Non-autoregressive Neural Machine Translation: Tricks of the Trade. In Findings of ACL
> - Qian et al., 2021. Glancing Transformer for Non-Autoregressive Neural Machine Translation. In ACL

---

### Official Review · Reviewer_byW1 · 2021-07-20

**Rating:** 7
**Confidence:** 5

**Summary:**

This paper proposes duplex sequence-to-sequence learning where data streams can flow from both ends of a seq2seq model. More precisely, they design a parameter efficient Transformer (REDER) that can simultaneously input and output a distinct language from each end of the model. In this way, REDER enables reversible machine translation by simply flipping the input and output ends. The experimental results show that the proposed REDER improve the multitask trained baseline by over 1.3 BLEU while maintaining over 5x speed-up at inference time.

**Limitations And Societal Impact:**

The authors claimed to have addressed the limitations. However they did not show that in a separate section or discussion, which makes the reviewer difficult to find it.

**Main Review:**

Overall, I think the idea of duplex learning is very interesting and novel. Different from standard multi-task learning, the proposed method shows evidence that it nicely combines the information from both directions, while avoiding hurting the performance due to the capacity issue of modeling two languages. Although the basic idea of reversible layers is borrowed from previous work, it is novel to me that the same idea can be applied to Transformers.

However, I still have the following questions:

(1) When applying upsampling to the source tokens, how can reversibility still be maintained? In Line 62, the definition means that you need to return to x when applying forward and backward functions to x. However, the CTC loss allows the model to predict any sentences with the same collapsed form. To rephrase my question, I think the collapse process used in CTC is not reversible, but I am not sure about that.

(2) For the cycle consistency loss (Line 189), is it differentiable to the forward pass? Since you need to generate target sentences first before sending it back to the backward pass, and generation is non-differentiable.

(3) How do you determine the coefficients of $\lambda_{cc}$ and $\lambda_{fba}$?

(4) The proposed REDER has a similar number of parameters compared to standard NAT models (e.g. Gu and Kong, 2020). However, the vanilla model of REDER still works worse than it does despite it utilizes both directions. What do you think may be the cause? Note that the results of Gu and Kong, 2020 in table 1 is greedy without using any reranking and beam-search, while the results of REDER does. It would be better to denote that in the table caption.

(5) Will the relative self-attention also benefit the standard simplex NAT models?

(6) The results without KD look quite low. Although it generally makes sense, what do you think how we can improve the performance without KD within the REDER framework?


**Time Spent Reviewing:**

3-4 hours

---

> ### Author Response · Authors · 2021-08-10
> **Response to Reviewer byW1**
>
> Thanks very much for your valuable comments!
>
> &nbsp;
>
> **Q1:** When applying upsampling to the source tokens, how can reversibility still be maintained? In Line 62, the definition means that you need to return to x when applying forward and backward functions to x. However, the CTC loss allows the model to predict any sentences with the same collapsed form. To rephrase my question, I think the collapse process used in CTC is not reversible, but I am not sure about that.
>
> **A1:** Sorry if we created any confusion. REDER is reversible between the input embedding and the output embedding (i.e. the hidden representations before softmax) and it is not exactly reversible between the source sentence and target sentence due to existing irreversible operations (e.g., CTC collapse process).
>
> To be clearer, considering $x$ and $y$ as a pair of source and target sentences, $emb(x^{upsampled})$ is the input embedding of the upsampled $x$ and $H^{L}$ is the output hidden representations of the network that correspond to the predicted CTC alignment $a$. REDER allows reversibility between $emb(x^{upsampled})$ and $H^{L}$ by its architecture design. However, we cannot assure the exact reversibility in the discrete token level beween $x$ and $y$, because of the irreversible argmax operation discretizing probabilities to tokens, as well as the irreversible CTC collapse process you mentioned.
>
> Empirically still, REDER shows a decent reconstruction capability in practice, where the BLEU score between source sentences and their round-trip reconstructions, i.e. $\text{BLEU}(x, f_\theta^{\leftarrow}f_\theta^{\rightarrow}(x))$, is 66.0 on WMT14 EN-DE. Also, a case of reconstruction has been shown in FIgure 2 in the Appendix.
>
> &nbsp;
>
> **Q2:** For the cycle consistency loss (Line 189), is it differentiable to the forward pass?
>
> **A2:** No, the cycle consistency is not differentiable to the forward pass In our current implementation. However, the cycle consistency loss could also improve the forward pass since the model parameters $\theta$ are shared between two directions.
>
> Thanks for your comment! A possible solution to make it differentiable to the forward pass could be introducing the Gumbel-softmax. We would leave this for further exploration.
>
> &nbsp;
>
> **Q3:** The proposed REDER has a similar number of parameters compared to standard NAT models (e.g. Gu and Kong, 2020). However, the vanilla model of REDER still works worse than it does despite it utilizes both directions. What do you think may be the cause?  Note that the results of Gu and Kong, 2020 in table 1 is greedy without using any reranking and beam-search, while the results of REDER does. It would be better to denote that in the table caption.
>
> **A3:** Sorry if we made any confusion.
>
> - **About the number of parameters**. To enable two translation directions, the standard NAT models (e.g., Gu and Kong, 2020) need 2 models. As a result, their parameters are roughly as double as REDER (124M vs 58M).
> - **About model performance**. The current SoTA NAT models in Gu and Kong (2020) were built by combing the advanced NAT modeling strategies proposed by different papers, e.g., GLAT, VAE. If using CTC only, the result of their model is 26.51, while REDER is 26.89 (using greedy decoding, in Table 3). This shows that REDER is also competitive when using greedy decoding and not using other advanced training techniques.\
> Furthermore, we believe that their best practice of training NAT models could also supplement to improving REDER. We will leave this for further study.
> - **About table caption**. We'll revise the caption of the table regarding your suggestions. Besides, Table 3 shows that REDER using greedy decoding archives 26.89 compared to 27.20 of CTC+GLAT. Meanwhile REDER obtains a faster decoding speedup as REDER does not consist of extra cross-attention layers.
>
> &nbsp;
>
> **Q4:** Will relative self-attention also benefit standard NAT models?
>
> **A4:** Yes, relative self-attention could also benefit standard NAT models, but not as much as for REDER. We conducted a quick experiment as you suggested. Relative self-attention improves a standard NAT model by 0.2 BLEU score on WMT14 En-De in our experiments.  On the contrary, as shown in Table 2, relative self-attention improves REDER by 0.54 BLEU score.
>
> To analyze, relative self-attention is known to better model reordering between consecutive layers. It is very important for REDER, which could be seen as an encoder-only model, to learn the reordering between input (source) and output (target), thus a better layer-wise reordering capability is needed. Therefore, relative self-attention is very helpful in this case. On the contrary, standard NAT models are of the encoder-decoder framework, the reordering between source and target sentences could be modeled by cross-attention. Hence relative self-attention would not bring significant benefits.
>
> &nbsp;
>
>
> **Q5:** What do you think how we can improve the performance without KD within the REDER framework?
>
> **A5:** Thanks for your questions. Please refer to the general response.
>
> &nbsp;
>
>
> **Q6:** How do you determine the coefficients of $\lambda_{fba}$ and $\lambda_{cc}$?
>
> **A6:** We determined each of them by searching in a set of {0.5, 0.25, 0.1, 0.05} and found that 0.1 worked the best. We did not try their combinations so 0.1 is not necessarily the best value.
>
> ---
> **References:**
> - Gu and Kong, 2021. Fully Non-autoregressive Neural Machine Translation: Tricks of the Trade. In Findings of ACL

---

### Author Response · Authors · 2021-08-10
**General Response**

We thank all reviewers for your insightful comments.  We try to address crucial common concerns as follows:

&nbsp;


**Q1:** The proposed REDER seems heavily relying on knowledge distillation data. What do you think how we can improve the performance without KD within the REDER framework?

**A1:** Yes, like other NAT approaches, the proposed REDER resort to KD data for training. This may hinder the generalization of NAT models including REDER to other applications especially multilingual scenarios. Here are our thoughts on this:

- **About the general KD requirement in NATs.** We notice that recent studies could have the potentials for the removal of KD for NAT models, through introducing latent-variable models (Gu and Kong, 2020; Bao et al 2021) or sampling/denoising-based augmented training objectives/strategies (Qian et al, 2020). As KD-dependence is a common issue for all NAT approaches, we believe future breakthroughs would resolve this. \
Please note that the aim of this paper is to make the idea of duplex sequence-to-sequence learning and its implementation of REDER realizable, at least in the scenario with KD data. Eliminating the need for KD is orthogonal to the purpose of this paper, however, is very valuable for further exploration.

- **Some possible ideas of eliminating the need for KD for REDER.**  As previous studies have pointed, KD plays a role in reducing the difficulty of training.  Inspired by the recent advances of pretraining models on monolingual data and finetuning for translation, one possible direction could be a better parameter-initialization of NAT models. Particularly for REDER, we are interested in seeing if using monolingual raw data to pretrain each "end" (i.e., the source/target half of a REDER) in a BERT-like MLM objective, then combining them as a whole REDER and continuing to train on raw parallel data could have any positive implications or insights.

Hopefully we could address your concerns! We will release our codes soon if accepted.


---
**References:**
- Gu and Kong, 2021. Fully Non-autoregressive Neural Machine Translation: Tricks of the Trade. In Findings of ACL
- Bao et al., 2021. Non-Autoregressive Translation by Learning Target Categorical Codes. In NAACL
- Qian et al., 2021. Glancing Transformer for Non-Autoregressive Neural Machine Translation. In ACL

---

### Decision · Program_Chairs · 2021-09-27

**Decision:**

Accept (Poster)

**Comment:**

The paper explores a "duplex" architecture for machine translation, by designing a network in which either end can be used as an input or output. To achieve this, the authors apply previous ideas about reversible networks to the transformer architecture, and use a non-autoregressive loss. The reviewers found this to be a novel, interesting and elegant approach. Like most non-autoregressive approaches, the need for knowledge distillation remains a limitation, and the authors reasonably point out that removing KD is not a goal of the paper. There are also questions about whether the approach really has large practical advantages over existing work - but this topic seems to be an interesting direction that is worth further exploration, so I recommend acceptance.